# Life-Threatening Hypokalemic Paralysis and Prevention of Severe Rebound Hyperkalemia in a Female with Barium Poisoning: A Rare Case Report

**DOI:** 10.3390/reports7030072

**Published:** 2024-08-28

**Authors:** Ting-Wei Liao, Ruei-Lin Wang, Szu-Chi Chen, Ya-Chieh Chang, Wen-Fang Chiang, Po-Jen Hsiao

**Affiliations:** 1Division of Medicine, Taoyuan Armed Forces General Hospital, Taoyuan 325, Taiwan; doc62388@aftygh.gov.tw (T.-W.L.); aassukw@livemail.tw (R.-L.W.); doc62606@aftygh.gov.tw (S.-C.C.); 2Division of Nephrology, Department of Internal Medicine, Taoyuan Armed Forces General Hospital Hsinchu Branch, Hsinchu 300, Taiwan; ajie1124@gmail.com; 3Division of Nephrology, Department of Internal Medicine, Tri-Service General Hospital, National Defense Medical Center, Taipei 114, Taiwan; wfc96076@aftygh.gov.tw; 4Division of Nephrology, Department of Internal Medicine, Taoyuan Armed Forces General Hospital, Taoyuan 325, Taiwan; 5Department of Life Sciences, National Central University, Taoyuan 320, Taiwan

**Keywords:** barium poisoning, hypokalemic paralysis, rebound hyperkalemia, respiratory failure, cardiac arrhythmias, potassium inward rectifier channel

## Abstract

Hypokalemic paralysis is a clinical syndrome characterized by acute flaccid paralysis with concomitant hypokalemia. Complications, such as acute respiratory failure and cardiac arrhythmias, can be fatal. If treated appropriately, the patient can recover without any sequelae. We present a rare case of life-threatening hypokalemic paralysis following the ingestion of an unknown substance. At presentation, her serum potassium concentration was 1.9 mmol/L. A review of the patient’s history confirmed the ingestion of barium chloride. She was diagnosed with acute barium poisoning characterized by high serum and urine barium levels. Aggressive potassium repletion was administered intravenously and orally. Her serum potassium concentration dropped to 1.5 mmol/L and peaked at 5.4 mmol/L following treatment. The patient achieved a complete recovery and was discharged without sequelae. Barium can competitively block the potassium inward rectifier channels and interfere with the efflux of intracellular potassium, leading to severe hypokalemia. Our report illustrates a rare presentation of acute barium intoxication and a differential diagnosis indicating hypokalemic paralysis. We also discuss the pathophysiological features and compare the clinical findings with cases of rebound hyperkalemia.

## 1. Introduction

Hypokalemic paralysis represents a heterogeneous group of disorders characterized by episodic attacks of acute flaccid paralysis, potentially leading to life-threatening complications such as acute respiratory failure and cardiac arrhythmias [1,2]. Primary hypokalemic paralysis, which may be familial with autosomal-dominant inheritance or sporadic, is a channelopathy caused by skeletal muscle ion channel mutations. Secondary causes include thyrotoxicosis, renal tubular acidosis, Gitelman syndrome, primary hyperaldosteronism, and gastrointestinal potassium losses [3,4,5]. In our case, barium intoxication is also a rare cause of hypokalemic paralysis.

Barium is a heavy alkaline earth metal found in barite (BaSO₄), witherite (BaCO₃), and other natural minerals, and it has a variety of uses in industry and medicine. The toxicity of barium is related to the solubility of the compound. Soluble barium compounds, such as those containing acetate, chloride, hydroxide, and nitrate, are toxic when ingested. Acute barium poisoning is rare and mostly results from food poisoning and contaminated contrast agents. Such poisoning can also cause secondary hypokalemic paralysis due to the competitive blockage of potassium inward rectifier channels (Kir) and interference with the efflux of intracellular potassium [6]. Indeed, complications such as acute respiratory failure and cardiac arrhythmias can be fatal [6,7]. Aggressive intravenous potassium repletion is necessary to reverse these toxic effects. Rebound hyperkalemia should be avoided with close follow-up of potassium concentration.

We present a case of hypokalemic paralysis caused by barium poisoning, which manifested as profound hypokalemia and acute flaccid paralysis. Hyperkalemia developed following potassium supplementation but was successfully managed to prevent further complications of rebound hyperkalemia.

## 2. Case Presentation

A 26-year-old female presented to our emergency department with nausea, vomiting, and four-limb paralysis after attempting suicide by ingesting unknown substances 2 h prior. Upon examination, she was afebrile, with a blood pressure of 121/86 mmHg, a pulse rate of 85 beats per minute, and a respiratory rate of 18 breaths per minute. Pulse oximetry revealed 97% oxygenation while breathing ambient air. She was awake and oriented (GCS score: E4M6V5) but presented muscle weakness with a power of 1/5 globally and diminished reflexes. Pupils were dilated with sluggish light reflexes. An electrocardiogram showed diffuse ST depression with flattened T waves (Figure 1). Biochemical analysis revealed severe hypokalemia at a concentration of 1.9 mmol/L, as detailed in Table 1.

Gastric lavage was performed, followed by the administration of activated charcoal. Potassium was delivered intravenously via a central line (10 mmol/h for 3 h), accompanied by an oral dose of 50 mmol. Following the initial management, the patient disclosed the ingestion of six pills of barium chloride in a suicide attempt. The pills were obtained from a chemical plant by her friend, with the dosage unknown. Otherwise, the patient denied any history of medication. Four hours after admission, her potassium concentration declined to 1.5 mmol/L, and respiratory failure developed (SpO2: 85% under a simple mask 10 L/min). We increased the potassium infusion rate (20 mmol/h for 4 h) and administered oxygen via a non-rebreathing mask (15 L/min). Her serum potassium concentration stabilized at 3.7 mmol/L after administering approximately 160 mmol of KCl over 8 h, accompanied by an improvement in respiratory failure. We discontinued potassium infusion immediately and adjusted oxygen support to a simple mask (10 L/min). Ten hours after admission, the patient’s potassium concentration increased to 5.4 mmol/L, accompanied by hyperacute T waves and widened QRS tachycardia on the electrocardiogram. Acute kidney injury was noted, with serum creatinine increasing from 0.93 to 1.38 mg/dL. After we corrected the hyperkalemia to 4.8 mmol/L using insulin and sodium bicarbonate within 2 h, the patient’s electrocardiogram returned to sinus rhythm. Toxicological analysis using inductively coupled plasma-atomic emission spectrometry (ICP-MS) confirmed barium intoxication, with blood and urine barium concentrations of 8769 μg/L and 11,426 μg/L, respectively (Table 1). Following medical treatment, the patient’s renal function improved, and her serum creatinine decreased to 0.26 mg/dL. After 8 days of hospitalization, the serum barium level decreased to 26 μg/L. She was discharged and achieved a complete recovery without any sequelae after 3 months of follow-up.

## 3. Discussion

Hypokalemic paralysis is a syndrome characterized by acute flaccid weakness that can be classified as primary (familial) or secondary, according to its etiology. Primary hypokalemic periodic paralysis is a channelopathy caused by skeletal muscle ion channel mutations affecting calcium or sodium channels. Secondary causes include thyrotoxicosis, renal tubular acidosis, Gitelman syndrome, primary hyperaldosteronism, and gastrointestinal potassium losses (Table 2) [3,4,5,8,9]. In rare cases, acute barium poisoning also causes secondary hypokalemic paralysis due to intracellular shifting. Hypokalemia predominantly affects the neuromuscular system, leading to muscle weakness, especially in the lower extremities. Severe cases can cause generalized muscle weakness, paralysis, respiratory failure, and arrhythmias [1,2,5]. Deep tendon reflexes are often diminished or absent, but sensation and consciousness usually remain unaffected. Potassium replacement is essential for the treatment. Further management depends on the etiology of hypokalemia [5].

Acute barium poisoning is rare and mostly results from food poisoning and contaminated contrast agents. The minimum oral toxic dose has been reported to be 200 mg, with lethal doses ranging from 1 g to 30 g [10]. Acute toxicity is characterized by its hypokalemic effect. Barium competitively blocks potassium inward rectifier channels (Kir) and interferes with the efflux of intracellular potassium without affecting the activity of the sodium–potassium ATPase pump (Figure 2) [6,11]. Thus, the continuous decrease in extracellular potassium leads to profound hypokalemia. Barium may enhance the permeability of the cell membrane to sodium, causing a secondary increase in Na^+^-K^+^ pump activity, which leads to a shift of extracellular potassium into the cell (Figure 2) [12,13]. It also mimics the physiological effects of calcium on nerve and muscle activity, leading to the release of neurotransmitters, such as acetylcholine, noradrenaline, and catecholamines, from nerve endings and the adrenal medulla [14]. Additionally, it was reported that barium chloride can cause direct injury to skeletal muscle through calcium-dependent proteolysis secondary to membrane depolarization [15]. As a result, barium intoxication results in potent muscular and cardiac toxicity.

Acute intoxication manifests within 10–60 min with severe gastrointestinal symptoms, such as vomiting, abdominal cramps, and diarrhea. The toxicity can progress within 2–4 h to cardiovascular and neuromuscular symptoms, leading to flaccid paralysis, areflexia, respiratory failure, hypertension and/or hypotension, and cardiac arrhythmias [6,7]. ECG abnormalities are common, with flattening and inversion of the T wave, prominent U wave, and ventricular arrhythmias. Only a positive toxicology test for barium can provide a definitive diagnosis. Higher barium concentrations do not result in a further dose-dependent decrease in plasma potassium concentration [6,7]. Human data on the renal toxicity of barium are limited, but cases of barium poisoning have reported acute renal failure as a complication [16]. Studies on reproductive and developmental effects are limited and inconclusive [16].

Initial treatment of barium intoxication includes oral administration of soluble sulfates to limit the absorption of barium by precipitating barium ions to insoluble, nontoxic barium sulfate [12]. The intravenous administration of sodium or magnesium sulfate is not recommended because barium sulfates may precipitate in the kidney, causing acute renal failure. Activated charcoal is ineffective since it cannot bind barium. For hypokalemia, treatment with aggressive intravenous potassium supplements is necessary to reverse the toxic effects. When the toxicity of barium is reversed, the blockade of potassium channels is also relieved with a reverse shifting of cellular potassium to the extracellular compartment. Serum potassium concentration should be monitored frequently during potassium infusion to prevent rebound hyperkalemia. The time to peak plasma levels is usually 2 h. Plasma barium levels decrease rapidly within 24 h. Renal elimination accounts for 10% to 28% of the total absorbed dose of barium, with the majority being eliminated through feces [12]. Hemodialysis may be beneficial for barium removal and the normalization of potassium [17]. The majority of patients recover without long-term sequelae. Most cases of death result from respiratory failure due to respiratory muscle paralysis and cardiac arrhythmias [6].

Our case represents a clinical manifestation of acute barium intoxication with profound hypokalemia and acute paralysis. The successful treatment and the benign outcome may be attributed to the prompt and substantial potassium supplementation. The patient’s initial severe hypokalemia was a direct result of barium toxicity. The decline in potassium concentration at 4 h might have resulted from the continuous effects of the toxin. Despite the presence of acidosis, the expected hyperkalemia was not observed, likely due to the underlying hypokalemia induced by barium. The mechanism by which barium causes hypokalemia may override the usual shift of potassium seen in acidosis, leading to a complex clinical picture where hypokalemia persists despite the acidosis.

Initially, the patient’s renal function was intact, but acute kidney injury developed after several hours, possibly due to the direct nephrotoxic effects of barium. Soluble sulfates were not administered before the onset of kidney injury in our patient. Barium has been reported to inhibit Kir channels in kidney cells, as shown in in vivo studies [18]. Kir channels are expressed in a variety of cell types, including those in the brain, heart, smooth and skeletal muscle, and kidney, where they contribute to diverse physiological functions, including the maintenance of resting membrane potential and the regulation of vascular smooth muscle tone. Kir2.1, expressed in the afferent and efferent arterioles of cortical nephrons, can be inhibited by barium, leading to vasoconstriction of the renal afferent arteriole and reduced blood flow. However, the effects and correlation with kidney damage require further investigation. Additionally, inhibition of Kir channels causes the contraction of vascular smooth muscle, which can lead to hypertension and reduced blood flow [19].

In our case, severe rebound hyperkalemia was prevented from causing serious complications after timely recognition and rapid correction. We have reviewed a few cases of rebound hyperkalemia in acute barium intoxication (Table 3) [20,21,22]. Rebound hyperkalemia is rarely reported and can develop in severe cases that require substantial potassium infusion. A total potassium supplementation of 80 mmol can cause rebound hyperkalemia. It is imperative for physicians in emergency departments to be aware of the life-threatening hypokalemic paralysis caused by barium poisoning and the potential complications of rebound hyperkalemia.

## 4. Conclusions

Acute barium poisoning is a rare and potentially fatal cause of hypokalemic paralysis. An extensive evaluation of the patient’s history is essential for determining the etiology of hypokalemic paralysis. Poisons and medications should also be reviewed, especially in cases involving the ingestion of unknown substances. Rapid correction of severe hypokalemia is crucial, and serum potassium concentration should be closely monitored to prevent rebound hyperkalemia. If recognized and treated appropriately, patients can recover without any clinical sequelae.

## Figures and Tables

**Figure 1 reports-07-00072-f001:**
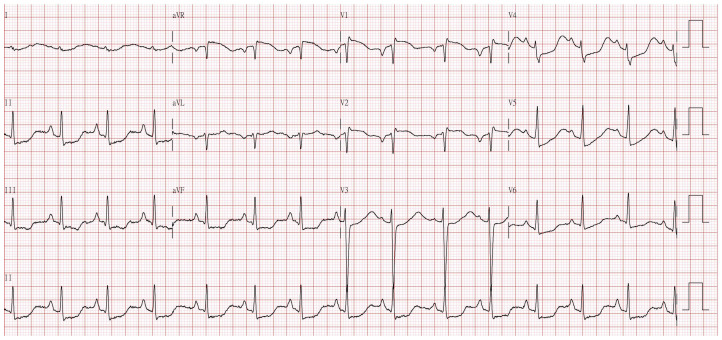
Electrocardiogram showed diffuse ST depression with flattened T waves and prominent U waves.

**Figure 2 reports-07-00072-f002:**
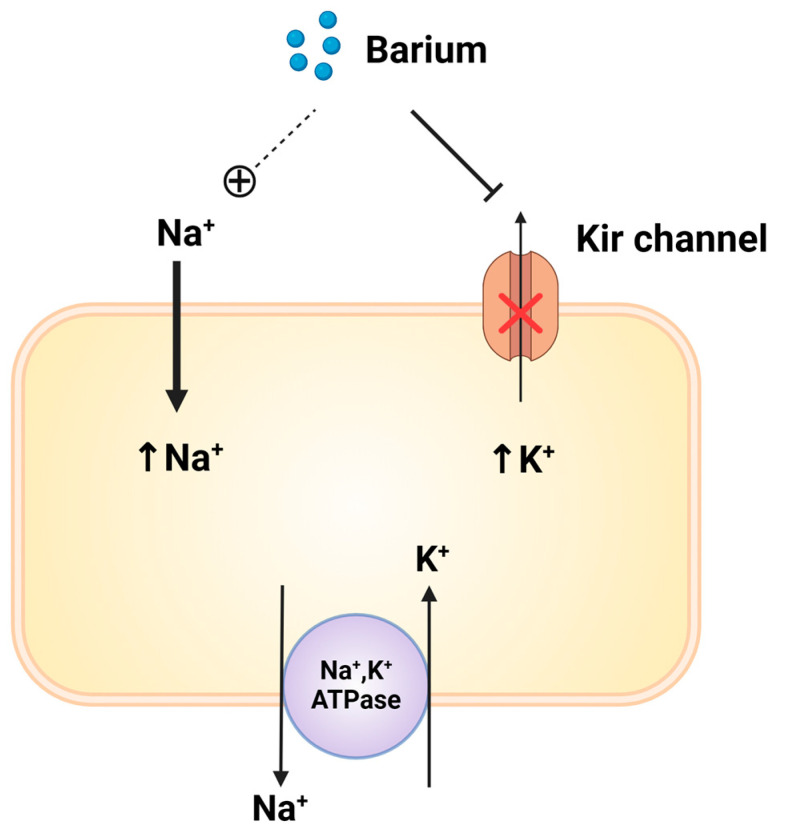
Mechanism of hypokalemia due to barium poisoning. 1. Barium directly blocks the outflow of potassium through the potassium inward rectifier channels (Kir). 2. Barium may enhance the permeability of the cell membrane to sodium, causing a secondary increase in Na^+^-K^+^ pump activity, which leads to a shift of extracellular potassium into the cell. This image was created with BioRender “https://biorender.com (Accessed on 20 August 2024)”.

**Table 1 reports-07-00072-t001:** Blood and urine biochemistry data during the first 12 h after admission to the emergency department.

Time (h)	0	4	8	10	12	Normal Value
**Serum**						
BUN	10.8	-	-	20.1	-	(6–20 mg/dL)
Creatinine	0.93	-	-	1.38	-	(0.5–0.9 mg/dL)
Sodium	137.5	139.1	137.8	145.4	147.7	(136–145 mmol/L)
Potassium	1.9	1.5	3.7	5.4	4.8	(3.5–5.1 mmol/L)
Magnesium	2.59	2.62	-	2.48	-	(1.8–2.55 mg/dL)
Calcium	8.67	-	-	8.58	-	(8.8–10.6 mg/dL)
Chloride	104	106.5	112	114	116.9	(99–107 mmol/L)
Phosphorus	3.45	4.76	3.47	-	-	(2.6–4.4 mg/dL)
CPK	129	100	92	-	-	(13–167 U/L)
Arterial pH	7.325	7.067	7.32	7.314	7.336	(7.35–7.45 mmHg)
PaCO_2_	43.3	69	29.9	39.4	41.6	(35–45 mmHg)
PaO_2_	90	78.2	245	133	199	(75–100 mmHg)
Bicarbonate	22.8	20	15.8	20.2	22.5	(22–26 mmol/L)
Anion gap (AG)	10.7	12.6	10	11.2	8.3	(8–12)
Barium		8769	-	-		(1–60 μg/L)
**Urine**						
Sodium	-	87.2	-	-	-	(mmol/L)
Potassium	-	22.1	-	-	-	(mmol/L)
Chloride	-	127.5	-	-	-	(mmol/L)
Protein	-	44.1	-	-	-	(mg/dL)
Creatinine	-	75.11	-	-	-	(mg/dL)
Potassium/creatinine	-	3.3	-	-	-	(<2 mmol/mmol)
Barium	-	11,426	-	-	-	(1.0–7.0 μg/L)

Anion Gap = [Na⁺] − ([Cl⁻] + [HCO_3_⁻]). BUN: Blood urea nitrogen. CPK: Creatine phospho-kinase.

**Table 2 reports-07-00072-t002:** Differential diagnosis of hypokalemic paralysis.

Transcellular distribution of potassium
Primary (familial) periodic paralysis Thyrotoxic periodic paralysis Barium poisoning
Actual potassium depletion
Renal loss
Renal tubular acidosis (proximal and distal) Drugs (e.g., diuretics) Sjögren’s syndrome Intrinsic transport defects (e.g., Bartter’s syndrome, Gitelman’s syndrome and Liddle’s syndrome) Medullary sponge kidney Fanconi’s syndrome Primary hyperaldosteronism Chronic toluene exposure
Extrarenal loss
Celiac disease Tropical sprue Acute gastroenteritis Short-bowel syndrome

**Table 3 reports-07-00072-t003:** Rebound hyperkalemia in acute barium intoxication.

Reports	Age/Sex	Complications	Initial Potassium Concentration	Total Potassium Dose (Hours)	Potassium Concentration after Treatment	Outcome
Wetherill SF (1981)	52/M	Paralysis, respiratory failure	2.9 mmol/L	240 mmol (16 h)	8.0 mmol/L	Recovered
Johnson CH (1991)	48/M	Paralysis, arrhythmia, respiratory failure	2.5 mmol/L	320 mmol (7 h)	7.5 mmol/L	Recovered
Johnson CH (1991)	38/F	Paralysis, respiratory failure	2.0 mmol/L	120 mmol (3 h)	6.3 mmol/L	Recovered
Sigue G (2000)	25/M	Paralysis, respiratory failure,	1.5 mmol/L	80 mmol (18 h)	8.2 mmol/L	Recovered
Our case	26/F	Paralysis, respiratory failure	1.9 mmol/L	160 mmol (8 h)	5.4 mmol/L	Recovered

## Data Availability

The data presented in this study are available on reasonable request from the corresponding author. The data are not publicly available due to privacy.

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
