# Peer review of "Life-Threatening Hypokalemic Paralysis and Prevention of Severe Rebound Hyperkalemia in a Female with Barium Poisoning: A Rare Case Report"

_reports, 2024, doi:10.3390/reports7030072_

Round 1

Reviewer 1 Report

Comments and Suggestions for Authors

Regarding the Case Report “Life-Threatening Hypokalemic Paralysis and Rebound Hyperkalemia in a Female with Barium Poisoning: A Rare Case Report”.

An unusual case of barium chloride poisoning (water-soluble salt) is reported in a 26-year-old woman who developed quadriparesis and respiratory failure caused by severe hypokalemia (1.5 mmol/L) after intentional ingestion.

There was an increase in kalemia values (5.4 mmol/L) with treatment and with the appearance of acute renal failure. There was an adequate therapeutic response with recovery ad integrum.

The case is well documented and easy to read.

Major observations

The mechanisms described for the development of hypokalemia are:

a. Competitive blockade of the potassium rectifier channel

b. Direct increase in cellular permeability to sodium, causing (via Na-K pump) displacement of potassium towards the intracellular.

The authors only emphasize the first of them.

In addition, direct action of barium on muscle and neuromuscular transmission have been described.

It is suggested that the authors develop these concepts in a little more detail.

Minor observations

Describe the pupillary examination: mydriasis?

Correct the Anion Gap formula indicated in Table 1.

Correct the abbreviation of pH and PaCO2.

Add phosphorus values in Table 1.

It is suggested that the discussion be eliminated regarding medications that can cause hypokalemia.

It is suggested that Table 3 be eliminated.

In the discussion, it is suggested that the discussion be eliminated regarding the carcinogenic role of barium.

Figure 1 can be improved and explained in more detail.

Author Response

Response to Reviewer 1

[General Comment]

An unusual case of barium chloride poisoning (water-soluble salt) is reported in a 26-year-old woman who developed quadriparesis and respiratory failure caused by severe hypokalemia (1.5 mmol/L) after intentional ingestion. There was an increase in kalemia values (5.4 mmol/L) with treatment and with the appearance of acute renal failure. There was an adequate therapeutic response with recovery ad integrum.The case is well documented and easy to read.

Author Reply: We sincerely appreciate your time and effort in reviewing this manuscript. We have thoroughly revised the manuscript according to all the reviewers' suggestions. Responses to your comments are provided below, and all changes have been highlighted in yellow.

Major comments:

The mechanisms described for the development of hypokalemia are:
a. Competitive blockade of the potassium rectifier channel
b. Direct increase in cellular permeability to sodium, causing (via Na-K pump) displacement of potassium towards the intracellular.
The authors only emphasize the first of them. In addition, direct action of barium on muscle and neuromuscular transmission have been described. It is suggested that the authors develop these concepts in a little more detail.

Author reply: We sincerely appreciate your time and effort spent reviewing this manuscript. We have revised the manuscript thoroughly according to your suggestions. Barium may enhance the permeability of the cell membrane to sodium, causing a secondary increase in Na+-K+ pump activity [12,13]. It also mimics physiological effects of calcium on nerve and muscle activity, leading to the release of neurotransmitters such as acetylcholine, noradrenaline, and catecholamines from nerve endings and the adrenal medulla [14]. Additionally, it was reported that barium chloride can cause direct injury to skeletal muscle through calcium-dependent proteolysis secondary to membrane depolarization [15]. As a result, barium intoxication results in potent muscular and cardiac toxicity. Please see the section of Discussion.

Minor comments:

  1. Describe the pupillary examination: mydriasis.

Author reply: We sincerely appreciate your valuable comment. We have added the pupilary examination in the revised manuscript. She was awake and oriented (GCS score: E4M6V5) but presented muscle weakness with a power of 1/5 globally and diminished reflexes. Pupils were dilated with sluggish light reflex. Please see the first paragraph of Case Presentation.

  1. Correct the Anion Gap formula indicated in Table 1

Author reply: We sincerely appreciate your valuable comments. We have corrected the Anion Gap formula in Table 1 as shown below.

Table 1. Blood and urine biochemistry data during the first 12 hours after admission to the emergency department.

Time (h)

0

4

8

10

12

Normal value

Serum

BUN

10.8

-

-

20.1

-

(6-20 mg/dL)

Creatinine

0.93

-

-

1.38

-

(0.5-0.9 mg/dL)

Sodium

137.5

139.1

137.8

145.4

147.7

(136-145 mmol/L)

Potassium

1.9

1.5

3.7

5.4

4.8

(3.5-5.1 mmol/L)

Magnesium

2.59

2.62

-

2.48

-

(1.8-2.55 mg/dL)

Calcium

8.67

-

-

8.58

-

(8.8-10.6 mg/dL)

Chloride

104

106.5

112

114

116.9

(99-107 mmol/L)

Phosphorus

3.45

4.76

3.47

-

-

(2.6-4.4 mg/dL)

CPK

129

100

92

-

-

(13-167 U/L)

Arterial pH

7.325

7.067

7.32

7.314

7.336

(7.35-7.45 mmHg)

PaCO₂

43.3

69

29.9

39.4

41.6

(35-45 mmHg)

PaO₂

90

78.2

245

133

199

(75-100 mmHg)

Bicarbonate

22.8

20

15.8

20.2

22.5

(22-26 mmol/L)

Anion gap (AG)

10.7

12.6

10

11.2

8.3

(8-12)

Barium

8769

-

-

(1-60 μg/L)

Urine

Sodium

-

87.2

-

-

-

(mmol/L)

Potassium

-

22.1

-

-

-

(mmol/L)

Chloride

-

127.5

-

-

-

(mmol/L)

Protein

-

44.1

-

-

-

(mg/dL)

Creatinine

-

75.11

-

-

-

(mg/dL)

Potassium/creatinine

-

3.3

-

-

-

(<2 mmol/mmol)

Barium

-

11426

-

-

-

(1.0–7.0 μg/L)

Anion Gap = [Na⁺] - ([Cl⁻] + [HCO₃⁻])

BUN: Blood urea nitrogen

CPK: Creatine phospho-kinase

  1. Correct the abbreviation of pH and PaCO2

Author reply: We sincerely appreciate your valuable comments. We have corrected the abbreviations of pH and PaCO2 in Table 1, as mentioned in minor comment #2.

  1. Add phosphorus values in Table 1.

Author reply: We sincerely appreciate your valuable comments. We have added the phosphorus values in Table 1, as mentioned in minor comment #2.

  1. It is suggested that the discussion be eliminated regarding medications that can cause hypokalemia.

Author reply: We sincerely appreciate your valuable comments. We have eliminated the discussion regarding medications that can cause hypokalemia. Please see the first paragraph of Discussion

  1. It is suggested that Table 3 be eliminated

Author reply: We sincerely appreciate your valuable comments. We have eliminated the Table 3 in the discussion.

  1. In the discussion, it is suggested that the discussion be eliminated regarding the carcinogenic role of barium.

Author reply: We sincerely appreciate your valuable comments. We have eliminated the discussion regarding the carcinogenic role of barium.

  1. Figure 1 can be improved and explained in more detail.

Author reply: We sincerely appreciate your valuable comments. We have revised Figure 1 from the original manuscript and replaced it with Figure 2 in the revised version, including detailed descriptions.

Barium competitively blocks potassium inward rectifier channels (Kir) and interferes with the efflux of intracellular potassium without affecting the activity of the sodium-potassium ATPase pump (Figure 2) [6,11]. Thus, the continuous decrease in extracellular potassium leads to profound hypokalemia. Barium may enhance the permeability of the cell membrane to sodium, causing a secondary increase in Na+-K+ pump activity, which leads to a shift of extracellular potassium into the cell (Figure 2) [12,13]. Please see the section of discussion.

.

Figure 2. Mechanism of hypokalemia due to barium poisoning. 1. Barium directly blocks the outflow of potassium through the potassium inward rectifier channels (Kir). 2. Barium may enhance the permeability of the cell membrane to sodium, causing a secondary increase in Na+-K+ pump activity, which leads to a shift of extracellular potassium into the cell.

Last, we are deeply honored by the time and effort you spent reviewing this manuscript. In reviewing and revising our manuscript, we are motivated to read more and thus learn more from your criticisms.

Reviewer 2 Report

Comments and Suggestions for Authors

Liao Barium poisoning Reports 20224

This is an interested case report describing the successful management of a barium poisoning incident in a young woman. The report is of interest clinically and with some slight improvements be beneficial for rapid treatment of such future cases. The report is well written and with appropriate details and conclusions.

Major

Title. The title seems a bit misleading which is to do with varying definitions of hyperkalemia, which differs between countries/regions. It currently also misses a potentially better title. Hyperkalemia can be defined as serum K+ concentration usually above 5 to 5.5 mM (NIH, USA); above 5.0 mM (European Society of Cardiology); above 5.5 mM (Australia) and above 5.5 mM (NHS). The patient after treatment had serum [K+] rise to 5.4 mM. This is defined as hyperkalemia in some of these settings and presumably also in Taiwan, but only by a small margin and for other regions, is not defined as hyperkalemia. To make this case report more internationally translatable, this title (and in report) should refer at most to only “mild” hyperkalemia. Even better would be to consider rewording the title to highlight the success of the treatment to “…. paralysis and prevention of severe rebound hyperkalemia….”.  This is consistent with comments of authors that rebound hyperkalemia is rarely reported and points to their improved means of managing barium poisoning.

Given the importance of the total K+ administered in this treatment efficacy, clearer and precise summary of K dosage, rate and length of infusion could be made in Section 2 Case Presentation. eg how long was initial central venous infusion? condense into 1-2 sentences suitable for busy clinicians to take away.

I like the clarity of Figure 1 but unfortunately it is too superficial to be useful as standalone figure; this can be easily improved with relatively minor changes. Figure 1 implies there is only a single K+ channel involved and only affecting efflux from the cell. Are authors referring to voltage activated K-channels (KIR, see Bhoelan cited 2014 paper) and/or KATP channels etc? Ditto only a single Na+ channel implied present in cell. Description is lacking from Figure caption – ie presumably this is a cell of some tissue, effects are based on what references etc.?

Minor

For precision, please refer to potassium concentration rather than potassium levels throughout the ms.

1.       Introduction.

last paragraph rewrite first sentence, with respect to (i) rebound hyperkalemia as per above criticism and (ii) indicating this occurred after treatment

2.       Case Presentation.

Patient PaO2was very low initially, which might be mentioned with SaO2 %. The value at 8 hours of 245 mmHg seems erroneous and should be discarded unless justified and demonstrated to be related to oxygen therapy. Value at 10 hours of 133 mmHg also seems very high given that PaCO2 appears normal? When was oxygen discontinued?

Add 12 hours to Table for at least K+ which shows final restoration (even if other measures cannot be fully detailed).

Did authors use any other K-lowering medications as eg in Table?

3.       Discussion.

A good reference indicating the independence of the Na+,K+-ATPase from K+ channels involved in barium inhibition is Clausen Overgaard J Physiology 527: 325-332, 2000.

Is it known what happens to endogenous catecholamines with barium poisoning?

A comment on the low pHa at onset effects on serum K+ would be an interesting inclusion.

I am unclear how barium was eventually removed in this patient?

Last paragraph, first sentence. Revise with respect to description of rebound hyperkalemia.

Comments on the Quality of English Language

Quality of written English was very good.

For precision state concentrations rather than levels.

Author Response

Response to Reviewer 2

[General Comment]

This is an interested case report describing the successful management of a barium poisoning incident in a young woman. The report is of interest clinically and with some slight improvements be beneficial for rapid treatment of such future cases. The report is well written and with appropriate details and conclusions.

Author Reply: We sincerely appreciate your time and effort in reviewing this manuscript. We have thoroughly revised the manuscript according to all the reviewers' suggestions. Responses to your comments are provided below, and all changes have been highlighted in yellow.

Major comments:

  1. The title seems a bit misleading which is to do with varying definitions of hyperkalemia, which differs between countries/regions. It currently also misses a potentially better title. Hyperkalemia can be defined as serum K+ concentration usually above 5 to 5.5 mM (NIH, USA); above 5.0 mM (European Society of Cardiology); above 5.5 mM (Australia) and above 5.5 mM (NHS). The patient after treatment had serum [K+] rise to 5.4 mM. This is defined as hyperkalemia in some of these settings and presumably also in Taiwan, but only by a small margin and for other regions, is not defined as hyperkalemia. To make this case report more internationally translatable, this title (and in report) should refer at most to only “mild” hyperkalemia. Even better would be to consider rewording the title to highlight the success of the treatment to “…. paralysis and prevention of severe rebound hyperkalemia….”. This is consistent with comments of authors that rebound hyperkalemia is rarely reported and points to their improved means of managing barium poisoning.

Author reply: We sincerely appreciate your time and effort spent reviewing this manuscript. We have revised the title “Life-Threatening Hypokalemic Paralysis and Prevention of Severe Rebound Hyperkalemia in a Female with Barium Poisoning: A Rare Case Report” according to your suggestions.

  1. Given the importance of the total K+ administered in this treatment efficacy, clearer and precise summary of K dosage, rate and length of infusion could be made in Section 2 Case Presentation. eg how long was initial central venous infusion? condense into 1-2 sentences suitable for busy clinicians to take away.

Author reply: We sincerely appreciate your valuable comment. We have revised the description about the potassium infusion dosage more precisely in case presentation as follows: Gastric lavage was performed, followed by the administration of activated charcoal. Potassium was delivered intravenously via a central line (10 mmol/hr for 3 hours), accompanied by an oral dose of 50 mmol. Following the initial management, the patient disclosed the ingestion of 6 pills of barium chloride in a suicide attempt. The pills were obtained from a chemical plant by her friend, with the dosage unknown. Four hours after admission, her potassium concentration declined to 1.5 mmol/L and respiratory failure developed (SpO2: 85% under simple mask 10L/min). We increased the potassium infusion rate (20 mmol/hr for 4 hours) and administered oxygen via a non-rebreathing mask (15L/min). Her serum potassium concentration stabilized at 3.7 mmol/L after the administration of approximately 160 mmol of KCl over 8 hours. Please see the section of Case Presentation.

  1. I like the clarity of Figure 1 but unfortunately it is too superficial to be useful as standalone figure; this can be easily improved with relatively minor changes. Figure 1 implies there is only a single K+ channel involved and only affecting efflux from the cell. Are authors referring to voltage activated K-channels (KIR, see Bhoelan cited 2014 paper) and/or KATP channels etc? Ditto only a single Na+ channel implied present in cell. Description is lacking from Figure caption – ie presumably this is a cell of some tissue, effects are based on what references etc.?

Author reply: We sincerely appreciate your valuable comments. We have revised Figure 1 from the original manuscript and replaced it with Figure 2 in the revised version, including detailed descriptions and reference as follows: Barium competitively blocks potassium inward rectifier channels (Kir) and interferes with the efflux of intracellular potassium without affecting the activity of the sodium-potassium ATPase pump (Figure 2) [6,11]. Thus, the continuous decrease in extracellular potassium leads to profound hypokalemia. Barium may enhance the permeability of the cell membrane to sodium, causing a secondary increase in Na+-K+ pump activity, which leads to a shift of extracellular potassium into the cell (Figure 2) [12,13]. It also mimics physiological effects of calcium on nerve and muscle activity, leading to the release of neurotransmitters such as acetylcholine, noradrenaline, and catecholamines from nerve endings and the adrenal medulla [14]. Additionally, it was reported that barium chloride can cause direct injury to skeletal muscle through calcium-dependent proteolysis secondary to membrane depolarization [15]. As a result, barium intoxication results in potent muscular and cardiac toxicity.

Figure 2. Mechanism of hypokalemia due to barium poisoning. 1. Barium directly blocks the outflow of potassium through the potassium inward rectifier channels (Kir). 2. Barium may enhance the permeability of the cell membrane to sodium, causing a secondary increase in Na+-K+ pump activity, which leads to a shift of extracellular potassium into the cell.

Minor comments:

  1. For precision, please refer to potassium concentration rather than potassium levels throughout the ms.

Author reply: We sincerely appreciate your valuable comments. We have revised the potassium concentration throughout the manuscript.  

  1. Introduction
    Last paragraph rewrite first sentence, with respect to (i) rebound hyperkalemia as per above criticism and (ii) indicating this occurred after treatment

Author reply: We sincerely appreciate your valuable comment. We have revised the introduction according to your suggestion as follows: We present a case of hypokalemic paralysis caused by barium poisoning, which manifested as profound hypokalemia and acute flaccid paralysis. Hyperkalemia developed following potassium supplementation but was successfully managed to prevent further complications of rebound hyperkalemia. Please see the last paragraph of Introduction

  1. Patient PaO2 was very low initially, which might be mentioned with SaO2 %. The value at 8 hours of 245 mmHg seems erroneous and should be discarded unless justified and demonstrated to be related to oxygen therapy. Value at 10 hours of 133 mmHg also seems very high given that PaCO2 appears normal? When was oxygen discontinued?

Author reply: We sincerely appreciate your valuable comment. We have corrected the data on the initial PaO2 in Table 1. The value at 8 hours was accurate, corresponding to the application of a non-rebreathing mask (15 L/min) for the patient. When the PaO2 at 8 hours reached 245 mmHg, indicating improvement in respiratory failure, we adjusted the oxygen support to a simple mask at 10 L/min. We also revised the manuscript to specify the exact time when the oxygen support was adjusted as follows:

Four hours after admission, her potassium concentration declined to 1.5 mmol/L and respiratory failure developed (SpO2: 85% under simple mask 10L/min). We increased the potassium infusion rate (20 mmol/hr for 4 hours) and administered oxygen via a non-rebreathing mask (15L/min). Her serum potassium concentration stabilized at 3.7 mmol/L after administering ap-proximately 160 mmol of KCl over 8 hours, accompanied by an improvement in respiratory failure. We discontinued potassium infusion immediately and adjusted oxygen support to simple mask (10L/min). Please see the second paragraph in the section of Case Presentation.

  1. Add 12 hours to Table for at least K+ which shows final restoration (even if other measures cannot be fully detailed).

Author reply: We sincerely appreciate your valuable comment. We have added the data at 12 hours to Table 1 according to your suggestion.

Table 1. Blood and urine biochemistry data during the first 12 hours after admission to the emergency department.

Time (h)

0

4

8

10

12

Normal value

Serum

BUN

10.8

-

-

20.1

-

(6-20 mg/dL)

Creatinine

0.93

-

-

1.38

-

(0.5-0.9 mg/dL)

Sodium

137.5

139.1

137.8

145.4

147.7

(136-145 mmol/L)

Potassium

1.9

1.5

3.7

5.4

4.8

(3.5-5.1 mmol/L)

Magnesium

2.59

2.62

-

2.48

-

(1.8-2.55 mg/dL)

Calcium

8.67

-

-

8.58

-

(8.8-10.6 mg/dL)

Chloride

104

106.5

112

114

116.9

(99-107 mmol/L)

Phosphorus

3.45

4.76

3.47

-

-

(2.6-4.4 mg/dL)

CPK

129

100

92

-

-

(13-167 U/L)

Arterial pH

7.325

7.067

7.32

7.314

7.336

(7.35-7.45 mmHg)

PaCO₂

43.3

69

29.9

39.4

41.6

(35-45 mmHg)

PaO₂

90

78.2

245

133

199

(75-100 mmHg)

Bicarbonate

22.8

20

15.8

20.2

22.5

(22-26 mmol/L)

Anion gap (AG)

10.7

12.6

10

11.2

8.3

(8-12)

Barium

8769

-

-

(1-60 μg/L)

Urine

Sodium

-

87.2

-

-

-

(mmol/L)

Potassium

-

22.1

-

-

-

(mmol/L)

Chloride

-

127.5

-

-

-

(mmol/L)

Protein

-

44.1

-

-

-

(mg/dL)

Creatinine

-

75.11

-

-

-

(mg/dL)

Potassium/creatinine

-

3.3

-

-

-

(<2 mmol/mmol)

Barium

-

11426

-

-

-

(1.0–7.0 μg/L)

  1. Did authors use any other K-lowering medications as eg in Table?

Author reply: We managed the rebound hyperkalemia using only insulin and sodium bicarbonate after the aggressive infusion of KCl. The patient also denied any history of medication except for ingesting barium chloride. We have revised the manuscript with medication we used. Please see the second paragraph in section of Case Presentation.

  1. A good reference indicating the independence of the Na+,K+-ATPase from K+ channels involved in barium inhibition is Clausen Overgaard J Physiology 527: 325-332, 2000.

Author reply: We sincerely appreciate your valuable reference which provides a more evidence-based point regarding the mechanism of barium intoxication. We have added the reference in our manuscript as follows: Acute barium poisoning is rare and mostly results from food poisoning and con-taminated contrast agents. The minimum oral toxic dose has been reported to be 200 mg, with lethal doses ranging from 1 g to 30 g [10]. Acute toxicity is characterized by its hypokalemic effect. Barium competitively blocks potassium inward rectifier channels (Kir) and interferes with the efflux of intracellular potassium without affecting the ac-tivity of the sodium-potassium ATPase pump (Figure 2) [6,11]. Thus, the continuous decrease in extracellular potassium leads to profound hypokalemia.

Reference:
11.Clausen T, Overgaard K. The role of K+ channels in the force recovery elicited by Na+-K+ pump stimulation in Ba2+-paralysed rat skeletal muscle. J Physiol. 2000; 527: 325-332.

  1. Is it known what happens to endogenous catecholamines with barium poisoning?

Author reply: We sincerely appreciate your valuable comments. Barium can mimic the physiological effects of calcium on nerve and muscle activity, leading to the release of neurotransmitters such as acetylcholine, noradrenaline, and catecholamines from nerve endings and the adrenal medulla. We have added the discussion into our manuscript as follows: Barium competitively blocks potassium inward rectifier channels (Kir) and interferes with the efflux of intracellular potassium without affecting the activity of the sodium-potassium ATPase pump directly (Figure 2) [6,11]. Thus, the continuous decrease in extracellular potassium leads to profound hypokalemia. Barium may enhance the permeability of the cell membrane to sodium, causing a secondary increase in Na+-K+ pump activity, which leads to a shift of extracellular potassium into the cell [12,13]. It also mimics physiological effects of calcium on nerve and muscle activity, leading to the release of neurotransmitters such as acetylcholine, noradrenaline, and catecholamines from nerve endings and the adrenal medulla [14]. Addition-ally, it was reported that barium chloride can cause direct injury to skeletal muscle through calcium-dependent proteolysis secondary to membrane depolarization [15]. As a result, barium intoxication results in potent muscular and cardiac toxicity. Please see line 125-128 in page 4.

  1. A comment on the low pHa at onset effects on serum K+ would be an interesting inclusion.

Author reply: We sincerely appreciate your valuable comments. The acidosis is known to result in hyperkalemia. We have added the discussion to our manuscript as follows: Our case represents a clinical manifestation of acute barium intoxication with profound hypokalemia and acute paralysis. The successful treatment and the benign outcome may be attributed to the prompt and substantial potassium supplementation. The patient’s initial severe hypokalemia was a direct result of barium toxicity. The decline of potassium concentration at four hours might have resulted from the continuous effects of the toxin. Despite the presence of acidosis, the expected hyperkalemia was not observed, likely due to the underlying hypokalemia induced by barium. The mechanism by which barium causes hypokalemia may override the usual shift of potassium seen in acidosis, leading to a complex clinical picture where hypokalemia persists despite the acidosis. Please see the line 170-173, page 6 in the section of Discussion.

  1. I am unclear how barium was eventually removed in this patient?

Author reply: We sincerely appreciate your valuable comments. The peak plasma levels of barium are usually 2 hours. Plasma barium levels decrease rapidly within 24 hours. Renal elimination accounts for 10% to 28% of the total absorbed dose of barium, with the majority being eliminated through feces. Please see line 158-161 in page 5.

  1. Last paragraph, first sentence. Revise with respect to description of rebound hyperkalemia.

Author reply: We sincerely appreciate your valuable comments. We have revised the description of rebound hyperkalemia according to your suggestion.

Last, we are deeply honored by the time and effort you spent reviewing this manuscript. In reviewing and revising our manuscript, we are motivated to read more and thus learn more from your criticisms.

Reviewer 3 Report

Comments and Suggestions for Authors

The case is interesting and well-described. However, the following issues need to be added.

1.    In this case, please identify the source of barium chloride. Which agent contained barium chloride? Was it a commercial product?

2.    Please include urine electrolyte data in Table 1.

3.    The mechanisms by which barium blocks potassium channels should be further discussed. Which potassium channels are affected by barium? There are various potassium channels along the nephron. Please include a discussion on whether the kidney is involved in the process of barium-induced hypokalemia.

Author Response

Response to Reviewer 3

[General Comment]

The case is interesting and well-described. However, the following issues need to be added.

Author Reply: We sincerely appreciate your time and effort in reviewing this manuscript. We have thoroughly revised the manuscript according to all the reviewers' suggestions. Responses to your comments are provided below, and all changes have been highlighted in yellow.

Minor comments:

  1. In this case, please identify the source of barium chloride. Which agent contained barium chloride? Was it a commercial product?

Author reply: We sincerely appreciate your valuable comments. We have corrected the source of barium chloride as follows: Following the initial management, the patient disclosed the ingestion of 6 pills of barium chloride in a suicide attempt. The pills were obtained from a chemical plant by her friend, with the dosage unknown. Please see the section of Case presentation.

  1. Please include urine electrolyte data in Table 1.

Author reply: We sincerely appreciate your valuable comments. We have added urine electrolyte data in Table 1.

Table 1. Blood and urine biochemistry data during the first 12 hours after admission to the emergency department.

Time (h)

0

4

8

10

12

Normal value

Serum

BUN

10.8

-

-

20.1

-

(6-20 mg/dL)

Creatinine

0.93

-

-

1.38

-

(0.5-0.9 mg/dL)

Sodium

137.5

139.1

137.8

145.4

147.7

(136-145 mmol/L)

Potassium

1.9

1.5

3.7

5.4

4.8

(3.5-5.1 mmol/L)

Magnesium

2.59

2.62

-

2.48

-

(1.8-2.55 mg/dL)

Calcium

8.67

-

-

8.58

-

(8.8-10.6 mg/dL)

Chloride

104

106.5

112

114

116.9

(99-107 mmol/L)

Phosphorus

3.45

4.76

3.47

-

-

(2.6-4.4 mg/dL)

CPK

129

100

92

-

-

(13-167 U/L)

Arterial pH

7.325

7.067

7.32

7.314

7.336

(7.35-7.45 mmHg)

PaCO₂

43.3

69

29.9

39.4

41.6

(35-45 mmHg)

PaO₂

90

78.2

245

133

199

(75-100 mmHg)

Bicarbonate

22.8

20

15.8

20.2

22.5

(22-26 mmol/L)

Anion gap (AG)

10.7

12.6

10

11.2

8.3

(8-12)

Barium

8769

-

-

(1-60 μg/L)

Urine

Sodium

-

87.2

-

-

-

(mmol/L)

Potassium

-

22.1

-

-

-

(mmol/L)

Chloride

-

127.5

-

-

-

(mmol/L)

Protein

-

44.1

-

-

-

(mg/dL)

Creatinine

-

75.11

-

-

-

(mg/dL)

Potassium/creatinine

-

3.3

-

-

-

(<2 mmol/mmol)

Barium

-

11426

-

-

-

(1.0–7.0 μg/L)

  1. The mechanisms by which barium blocks potassium channels should be further discussed. Which potassium channels are affected by barium? There are various potassium channels along the nephron. Please include a discussion on whether the kidney is involved in the process of barium-induced hypokalemia.

Author reply: We sincerely appreciate your valuable comments. We have included the discussion as follows: Initially, the patient's renal function was intact, but acute kidney injury developed after several hours, possibly due to the direct nephrotoxic effects of barium. Soluble sulfates were not administered before the onset of kidney injury in our patient. Barium has been reported to inhibit Kir channels in kidney cells, as shown in in vivo studies [18]. Kir channels are expressed in a variety of cell types, including those in the brain, heart, smooth and skeletal muscle, and kidney, where they contribute to diverse physiological functions, including the maintenance of resting membrane potential and the regulation of vascular smooth muscle tone. Kir2.1, expressed in the afferent and efferent arterioles of cortical nephrons, can be inhibited by barium, leading to vasoconstriction of the renal afferent arteriole and reduced blood flow. However, the effects and correlation with kidney damage require further investigation. Additionally, inhibition of Kir channels causes the contraction of vascular smooth muscle, which can lead to hypertension and reduced blood flow [19]. Please refer to lines 174-186 in the discussion section.

Last, we are deeply honored by the time and effort you spent reviewing this manuscript. In reviewing and revising our manuscript, we are motivated to read more and thus learn more from your criticisms.

Reviewer 4 Report

Comments and Suggestions for Authors

What was the amount of Barium ingested?

At the value of 1.5 mmol of potassium, how did it not start with 20 mmol/hour? (being severe hypokalaemia)

You should add more keywords

What GCS did the patient have?

What other medications has the patient received? Apart from potassium

Did the patient's breathing deteriorate after receiving oxygen at 15l/min? Upon arrival, the patient's saturation was 97%

Why was the magnesium not measured at 4-8-10 hours? Hypokalemia is very often associated with hypomagnesemia. 

How many days was the patient hospitalized?

Don't you have an EKG of the patient?

Comments on the Quality of English Language

Minor editing English

Author Response

Response to Reviewer 4

We sincerely appreciate your time and effort in reviewing this manuscript. We have thoroughly revised the manuscript according to all the reviewers' suggestions. Responses to your comments are provided below, and all changes have been highlighted in yellow.

Comments:

  1. What was the amount of Barium ingested?

Author reply: We sincerely appreciate your valuable comments. We have revised the description of the barium chloride ingested to emphasize this point. The dosage was unknown in pills. Following the initial management, the patient disclosed the ingestion of 6 pills of barium chloride in a suicide attempt. The pills were obtained from a chemical plant by her friend, with the dosage unknown. Please see the section of Case presentation.

  1. At the value of 1.5 mmol of potassium, how did it not start with 20 mmol/hour? (being severe hypokalemia)

Author reply: We sincerely appreciate your valuable comment. We initiated the potassium infusion at 10 mmol/hr for preventing the rebound hyperkalemia. Additional oral potassium was also administered.

  1. You should add more keywords

Author reply: We sincerely appreciate your valuable comment. We have added key words according to your suggestion as follows:
Keywords: Barium poisoning; hypokalemic paralysis; rebound hyperkalemia; respiratory failure; cardiac arrhythmias; potassium inward rectifier channel 

  1. What GCS did the patient have?

Author reply: We sincerely appreciate your valuable comments. The patient was awake and oriented (GCS score: E4M6V5) upon arrival. We have added the GCS score to emphasize the point according to your suggestion. Please see the section of Case Presentation

  1. What other medications has the patient received? Apart from potassium

Author reply: We sincerely appreciate your valuable comments. We used activated charcoal for gastric lavage and primarily KCl for treating hypokalemia. Insulin and sodium bicarbonate were given to lower hyperkalemia after the potassium infusion. We revised the manuscript according to your point in the section of case presentation.

  1. Did the patient's breathing deteriorate after receiving oxygen at 15l/min? Upon arrival, the patient's saturation was 97%.

Author reply: We sincerely appreciate your valuable comment. The patient’s breathing did not deteriorate under oxygen at 15L/min which was shown in the arterial blood analysis at 8 hours (PaO2: 245mmHg). We adjusted the oxygen to 10L/min for the improving respiratory condition.

We have revised the description to make it more clear in the manuscript.

Gastric lavage was performed, followed by the administration of activated charcoal. Potassium was delivered intravenously via a central line (10 mmol/hr for 3 hours), accompanied by an oral dose of 50 mmol. Following the initial management, the patient disclosed the ingestion of 6 pills of barium chloride in a suicide attempt. The pills were obtained from a chemical plant by her friend, with the dosage unknown. Other-wise, the patient denied any history of medication. Four hours after admission, her potassium concentration declined to 1.5 mmol/L and respiratory failure developed (SpO2: 85% under simple mask 10L/min). We increased the potassium infusion rate (20 mmol/hr for 4 hours) and administered oxygen via a non-rebreathing mask (15L/min). Her serum potassium concentration stabilized at 3.7 mmol/L after administering ap-proximately 160 mmol of KCl over 8 hours, accompanied by an improvement in respiratory failure. We discontinued potassium infusion immediately and adjusted oxy\gen support to simple mask (10L/min). Please see line 77-84 in page 2.

  1. Why was the magnesium not measured at 4-8-10 hours? Hypokalemia is very often associated with hypomagnesemia. 

Author reply: We sincerely appreciate your valuable comments. We have added the data of magnesium in Table 1 according to your suggestion.

Table 1. Blood and urine biochemistry data during the first 12 hours after admission to the emergency department.

Time (h)

0

4

8

10

12

Normal value

Serum

BUN

10.8

-

-

20.1

-

(6-20 mg/dL)

Creatinine

0.93

-

-

1.38

-

(0.5-0.9 mg/dL)

Sodium

137.5

139.1

137.8

145.4

147.7

(136-145 mmol/L)

Potassium

1.9

1.5

3.7

5.4

4.8

(3.5-5.1 mmol/L)

Magnesium

2.59

2.62

-

2.48

-

(1.8-2.55 mg/dL)

Calcium

8.67

-

-

8.58

-

(8.8-10.6 mg/dL)

Chloride

104

106.5

112

114

116.9

(99-107 mmol/L)

Phosphorus

3.45

4.76

3.47

-

-

(2.6-4.4 mg/dL)

CPK

129

100

92

-

-

(13-167 U/L)

Arterial pH

7.325

7.067

7.32

7.314

7.336

(7.35-7.45 mmHg)

PaCO₂

43.3

69

29.9

39.4

41.6

(35-45 mmHg)

PaO₂

90

78.2

245

133

199

(75-100 mmHg)

Bicarbonate

22.8

20

15.8

20.2

22.5

(22-26 mmol/L)

Anion gap (AG)

10.7

12.6

10

11.2

8.3

(8-12)

Barium

8769

-

-

(1-60 μg/L)

Urine

Sodium

-

87.2

-

-

-

(mmol/L)

Potassium

-

22.1

-

-

-

(mmol/L)

Chloride

-

127.5

-

-

-

(mmol/L)

Protein

-

44.1

-

-

-

(mg/dL)

Creatinine

-

75.11

-

-

-

(mg/dL)

Potassium/creatinine

-

3.3

-

-

-

(<2 mmol/mmol)

Barium

-

11426

-

-

-

(1.0–7.0 μg/L)

  1. How many days was the patient hospitalized?

Author reply: We sincerely appreciate your valuable comments. The patient was hospitalized for 8 days. We have emphasized the point as follows: Following medical treatment, the patient’s renal function improved, and her serum creatinine decreased to 0.26 mg/dL. After eight days of hospitalization, the serum barium level decreased to 26 μg/L. She was discharged and achieved a complete recovery without any sequelae after 3 months of follow-up. Please see the section of Case Presentation.

  1. Don't you have an EKG of the patient?

Author reply: We sincerely appreciate your valuable comments. We have added the EKG at presentation as Figure 1 according to your suggestion.

Figure 1. Electrocardiogram showed diffuse ST depression with flattened T waves, and prominent U waves

Last, we are deeply honored by the time and effort you spent reviewing this manuscript. In reviewing and revising our manuscript, we are motivated to read more and thus learn more from your criticisms.

Round 2

Reviewer 1 Report

Comments and Suggestions for Authors

All modifications have been made satisfactorily, significantly improving the quality of the manuscript.

Reviewer 2 Report

Comments and Suggestions for Authors

Thank you for amending the manuscript. All my criticisms have been satisfactorily addressed. I appreciated the respectful nature of the response. For future reference, no need to include this in every response, just the once was sufficient.

Reviewer 3 Report

Comments and Suggestions for Authors

The authors properly addressed the questions.

I have no further comments.

Comments on the Quality of English Language

Fine

Reviewer 4 Report

Comments and Suggestions for Authors

Accept in present form